# Key Topics in Molecular Docking for Drug Design

**DOI:** 10.3390/ijms20184574

**Published:** 2019-09-15

**Authors:** Pedro H. M. Torres, Ana C. R. Sodero, Paula Jofily, Floriano P. Silva-Jr

**Affiliations:** 1Department of Biochemistry, University of Cambridge, Cambridge CB2 1GA, UK; monteirotorres@gmail.com; 2Department of Drugs and Medicines; School of Pharmacy; Federal University of Rio de Janeiro, Rio de Janeiro 21949-900, RJ, Brazil; acrsodero@gmail.com; 3Laboratório de Modelagem e Dinâmica Molecular, Instituto de Biofísica Carlos Chagas Filho, Universidade Federal do Rio de Janeiro, Rio de Janeiro 21949-900, RJ, Brazil; paula.jofily@gmail.com; 4Laboratório de Bioquímica Experimental e Computacional de Fármacos, Instituto Oswaldo Cruz, FIOCRUZ, Rio de Janeiro 21949-900, RJ, Brazil

**Keywords:** computer-aided drug design, structure-based drug design, benchmarking sets, consensus methods, fragment-based, machine learning

## Abstract

Molecular docking has been widely employed as a fast and inexpensive technique in the past decades, both in academic and industrial settings. Although this discipline has now had enough time to consolidate, many aspects remain challenging and there is still not a straightforward and accurate route to readily pinpoint true ligands among a set of molecules, nor to identify with precision the correct ligand conformation within the binding pocket of a given target molecule. Nevertheless, new approaches continue to be developed and the volume of published works grows at a rapid pace. In this review, we present an overview of the method and attempt to summarise recent developments regarding four main aspects of molecular docking approaches: (i) the available benchmarking sets, highlighting their advantages and caveats, (ii) the advances in consensus methods, (iii) recent algorithms and applications using fragment-based approaches, and (iv) the use of machine learning algorithms in molecular docking. These recent developments incrementally contribute to an increase in accuracy and are expected, given time, and together with advances in computing power and hardware capability, to eventually accomplish the full potential of this area.

## 1. Introduction

Molecular docking is a method which analyses the conformation and orientation (referred together as the “pose”) of molecules into the binding site of a macromolecular target. Searching algorithms generate possible poses, which are ranked by scoring functions [1]. Several software were developed during the last decades, amongst which are some well-known examples, such as AutoDock [2], AutoDock Vina [3], DockThor [4,5], GOLD [6,7], FlexX [8] and Molegro Virtual Docker [9].

The first step in a docking calculation is to obtain the target structure, which commonly consists of a large biological molecule (protein, DNA or RNA) [10] (Figure 1). The structures of these macromolecules can be readily retrieved from the Protein Data Bank (PDB) [11], which provides access to 3D atomic coordinates obtained by experimental methods. However, it is not unusual that the experimental 3D structure of the target is not available. In order to overcome this issue, computational prediction methods, such as comparative and ab initio modelling can be used to obtain the three-dimensional structure of proteins [1].

Usually, the binding site location on which to focus the docking calculations is known. However, when the binding region information is missing, there are two commonly employed approaches: either the most probable binding sites are algorithmically predicted or a “blind docking” simulation is carried out. The latter has a high computational cost, since the search covers all the target structure [12]. Several available software can be used to detect binding sites. MolDock [9], for example, uses an integrated cavity detection algorithm to identify potential binding sites. DoGSiteScorer is an algorithm that determines possible pockets and their druggability scores, which describe the potential of the binding site to interact with a small drug-like molecule [13]. Fragment Hotspot Maps [14] uses small molecular probes to identify surface regions in the receptor that are prone to interact with small molecules. These predicted interaction sites can then be provided as the centre of the sampling space.

Moreover, information derived from such hotspots or even from previous experimental knowledge (e.g., NMR, mass spectrometry) can be used to generate distance restraints, which is known to greatly increase protein-small molecule docking accuracy [15]. 

During docking calculations, a common strategy is to employ a grid representation that includes precalculated potential energies for interaction within the target binding site [16]. This approach speeds up the docking runs and basically consists of the discretisation of the binding site [17]. Then, at each grid point, interactions related to the Lennard–Jones and electrostatic potentials are calculated.

Ligand structure is also required and can be obtained from small molecules databases, such as ZINC [18] and PubChem [19]. These online databases facilitate the retrieval of a large number of compounds for subsequent virtual screening. If not directly available, the 3D atomic coordinates of these compounds can be obtained from the 2D structures (or even from simpler representation schemes, such as SMILES) using several available software, such as ChemSketch (Advanced Chemistry Development, Inc., Toronto, On, Canada, www.acdlabs.com, 2019), ChemDraw (PerkinElmer Informatics), Avogadro [20] and Concord [21]. It is worth noting that for small molecule ligands all that is needed initially is a stereochemically defined geometry with the correct relevant protonation state, since conformations will be explored by the docking software in the context of the target’s binding site.

Charges are usually assigned through algorithms that distribute the net charge of a molecule among its constituent atoms as partial atom-centred charges. Furthermore, most docking methods assume that a particular protonation state and charge distribution in the molecules do not change between their bound and unbound states [3]. Nevertheless, it is crucial for successful docking to evaluate free torsions, protonation states and charge assignments. The protonation states of the target’s amino acid residues can be critical to ligand interactions and, consequently, to the binding affinity prediction. There are several software available to evaluate the pKa of the amino acid residues, such as PropKa [22] and H++ [23]. 

Ligand protonation is also important since it affects the net charge of the molecule and the partial charges of individual atoms. Nonetheless, each docking program will employ a different charge assignment protocol [1]. For example, in the MolDock program, the protein and the ligands are automatically prepared (charges and protonation states assigned) and simplified charge and protonation schemes are used, as described by Thomsen and Christensen (2006). AutoDock uses Gasteiger–Marsili atomic charges whereas the closely-related AutoDock Vina does not require the assignment of atomic charges, since the terms that compose its scoring function are charge-independent [3,24]. The DockThor algorithm, as implemented in the homonymous web portal, automatically generates the topology files (i.e., atom types and partial charges) for the protein, ligand and cofactors according to the MMFF94S force field [4,5,25].

Two aspects are crucial to docking programs: search algorithms and scoring functions. The search algorithm analyses and generates ligand pose at a target’s binding site, taking into consideration the roto-translational and internal degrees of freedom of the ligand [10]. 

Search strategies are often classified as systematic, stochastic or deterministic [16]. Systematic search algorithms explore each ligand’s degree of freedom incrementally. As the number of free rotatable bonds increases, the number of evaluations can undergo a combinatorial explosion [16,26,27]. This class of search algorithms can be subdivided in exhaustive, incremental construction (which relies on the fragmentation of the ligand) and conformational ensemble [26]. FlexX [8] and eHits [28], for example, employ fragment-based approaches with systematic algorithms (incremental construction and graph matching, respectively).

A number of algorithms were also developed to use information from protein and ligand pharmacophores. Those algorithms try to match the distances between each of the ligand’s and protein’s pharmacophoric points [29]. The software FLEXX-PHARM, for example, is an extended version of FLEXX and applies pharmacophoric features as constraints into a docking calculation [30].

Stochastic search algorithms perform random changes in the ligand’s degrees of freedom. However, this kind of algorithm does not guarantee convergence to the best solution. To improve it, an iterative process can be performed. Monte Carlo, Evolutionary Algorithms (including genetic), Tabu Search and Swarm Optimisation are some of the most common stochastic algorithm implementations [26]. Several software use stochastic algorithms as search methods, such as AutoDock [2], GOLD [6], DockThor [4,5,25] and MolDock [9] (Table 1). 

In deterministic search, the orientation and conformation of the ligand in each iteration is determined by the previous state, and the new state has equal or lower energy value than the previous one [16,26]. However, this kind of algorithm has higher computational cost and often leads to the undesired trapping of the resulting conformations to a local energy minimum [16]. Examples are energy minimisation methods and molecular dynamics (MD) simulations. 

The overall size of the ligand, especially if it contains a large number of rotatable bonds impacts most docking algorithms in a negative way, both in terms of computational cost of each individual docking run and in terms of docking accuracy [46]. That is the case because each new rotatable bond inherently increases the ligand’s degrees of freedom, thus increasing the number of possible conformations. The enhanced conformational space is therefore much more complex to explore, rendering less accurate results, usually even with increased sampling steps. The magnitude of this effect is distinct for different algorithms [3,47] and fragment-based ones seem to exhibit superior performance in such cases [46].

Some algorithms can combine different search strategies, and often MD simulations are used to analyse the time-resolved trajectory of the ligand-bound system and to further pinpoint the best docking solutions [48,49,50,51].

After the generation of thousands of ligand orientations, additional scoring functions may be used to rank the conformations. They may be based on binding energy, free energy, or a qualitative numerical measure to approximate interaction energies [52]. Currently, scoring functions are grouped into three major types: force field, empirical and knowledge-based [26,27,53]. 

Force field-based functions consist of a sum of energy terms [26]. The potential energy usually accounts for bonded (bond length, angle, dihedrals) and nonbonded (van der Waals, electrostatic) terms. This type of function usually neglects solvent effects and entropies [16]. The DockThor program [4], for example, employs a scoring function for pose prediction based on the MMFF94S force field composed of three energy terms [54], i.e., the torsional term for bonded interactions, the electrostatic potential and the Buffered-14-7 term for the van der Waals potential (Equation (1)):(1)$$\begin{array}{l} E_{MMFF94}=\sum 0.5\left(V_{1}\left(1+\cos\Phi \right)+V_{2}\left(1-\cos2\Phi \right)+V_{3}\left(1+\cos3\Phi \right)\right)\\ \;+\sum \frac{332.0716q_{i}q_{j}}{\varepsilon \left(R_{ij}+\delta_{elec}\right)}+\sum \varepsilon_{ij}{\left(\frac{\left(1+\delta_{vdW}\right)R_{ij}^{*}}{R_{ij}+\delta_{vdW}R_{ij}^{*}}\right)}^{7}\left(\frac{\left(1+\gamma \right)R_{ij}^{*7}}{R_{ij}^{7}+\gamma R_{ij}^{*7}}-2\right), \end{array}$$
where V1, V2 and V3 are constants dependent on the types of the atoms *i* and *j*, *ϕ* is the i-j-k-l torsion angle, *q_i_* and *q_j_* are the partial charges of atoms *i* and *j*, *ε* is the dielectric constant given by a distance-dependent sigmoidal dielectric function [55], *R_ij_* is the internuclear separation between atoms *i* and *j*, and *δ_elec_* is the electrostatic buffering. Repulsion at short distances and van der Waals interactions are calculated by the last term, the Buf-14-7 potential [56]. In this term, $\varepsilon_{ij}$ is the well depth, $R_{ij}^{*}$ is the minimum-energy separation (Å) that depends on the MMFF94S types of the atoms *i* and *j*, and $\delta_{vdW}=0.07$ and γ=0.12 are the buffering constants.

Empirical scoring functions are derived from quantitative structure-activity relationships which were first idealised by Hansh and Fujita [16,57]. The goal is to predict binding affinity with high accuracy by using known experimental binding affinity data [26]. ChemScore [58] and GlideScore [59] are examples of empirical scoring functions.

Knowledge-based functions are based on frequency of atom pairs interactions observed in experimentally determined 3D structures of ligand-target complexes [16,26]. DrugScore of FlexX program [60] and PMF [61] are examples of knowledge-based functions.

Binding affinity prediction is still a major challenge for docking programs and most approaches rely upon consensus scoring schemes and rescoring approaches [16,26,27]. Consensus scoring for improving molecular docking accuracy is an ever-evolving research topic and will be addressed further in this review. 

### Molecular Docking in Drug Design

Molecular docking is a key component of the Computer-aided Drug Design toolbox. It is part of the so-called “structure-based drug design” methods and was first developed in the middle 80s through early 90s for predicting the binding mode of known active compounds and virtually screening large digital compound libraries to reduce costs and speed up drug discovery [62]. Docking tools have also been used in the hit-to-lead optimisation process. The latter application imposes the biggest challenge as predicting relative binding affinities for a series of related compounds has been the Achilles heel of most docking software since the very beginning of their development. Nevertheless, docking can still be used in hit-to-lead optimisation by indicating if the designed analogues of a hit compound present improved molecular interactions with the target.

Another widely known shortcoming of traditional docking methodologies is the poor modelling of receptor flexibility [63,64,65]. Some docking algorithms are able to partially mitigate this issue by allowing side-chain movement of active-site residues. Nevertheless, larger conformational changes might be triggered upon ligand-binding or might be a prerequisite to the binding event itself. A strategy, usually referred to as Receptor Ensemble Docking (or simply Ensemble Docking) is the most frequently used to model those scenarios. It is based on the concept of Conformational Selection and consists in using multiple conformations of the receptor molecule, that can be obtained via different methods, such as MD simulations [66,67], Normal Mode Analysis [68], and even by using alternative experimentally-determined receptor conformations [69]. It is worth noting that some software, such as GOLD and Glide have implemented functionality to execute this type of analysis.

The main limitations and challenges in the docking methodology have been identified nearly two decades ago [16] but they are still the subject of a very active research field. As described earlier here, two key components of the docking methodology are the conformational search algorithm and the scoring function. The former can suffer dramatically in performance when dealing with longer and flexible ligands, especially for shallow and chemically featureless binding sites, such as in polymer binding proteins (e.g., peptidases and glycosidases). Force-field based scoring functions suffer from the inherent problem of calculating binding affinities from the simplified interaction energies necessary to keep the docking calculations fast enough to process large compound libraries. Although binding affinities can be more accurately predicted from calculated binding free energies the latter also suffers from a problem of subtraction of large numbers (interaction energy between the ligand and protein on one hand and the cost of bringing the two molecules out of solvent and into an intimate complex on the other hand), which are often calculated with sub-optimal accuracy, and yield a small number as a result of the calculation [70]. 

In the following sections, we will review and discuss a selection of the main topics in the literature for molecular docking in drug design, all of which intend to address the above discussed limitations and advances in the methodology.

## 2. Benchmarking Sets

When using computational methods for molecular docking, it is paramount to assess the performance and accuracy of the programs to be employed. This not only allows one to know the degree of credibility that can be expected in the results, but also helps choosing the method or program better suited to the task at hand. To that end, there are many benchmarking databases that provide targets and ligands for docking, along with additional information such as true binding affinity, experimental binding pose, and actives/inactives distinction. Experimental information can then be compared to the docking program’s predictions through different statistical metrics, which allows the assessment of its performance.

### 2.1. Benchmarking Sets for Pose Prediction and Binding Affinity Calculations

The development of either empirical parametric or nonparametric regression models for docking pose and binding affinity predictions must be based on experimental data so that their functions may be properly parameterised (or inferred) and thus better represent reality. Moreover, the performance of these models must also be evaluated on such data. In light of this demand, there are many benchmarking datasets which aim to group as much high-quality data as possible [71,72,73,74].

The most widely employed of these is PDBBind [71]. This database is a result of an effort to screen the entire Protein Data Bank (PDB) [11] for experimentally determined 3D structures of protein-ligand complexes and collect their experimentally measured binding affinities. There is also a refined set of complexes [75] and a core set derived from it [76], which has become the standard set for benchmarking scoring functions (SFs). It is noteworthy that PDBBind is also widely used in training machine learning SFs for binding affinity predictions [77,78,79].

There are also benchmarking databases which encompass specific complexes or purposes, such as protein-protein complexes [80], membrane protein-protein complexes [81], and a blind set based on PDBBind for testing machine learning SFs [82].

Accuracy for pose prediction can be assessed by root mean square deviation (RMSD) calculations comparing predicted pose and experimental pose. To compare binding affinity predictions with experimentally determined affinities for a set of multiple data points, one can too calculate RMSD for the values, but also the Pearson correlation coefficient (R_p_) and the Spearman rank-correlation (R_s_) [83].

### 2.2. Benchmarking Sets for Virtual Screening

Benchmarking databases for virtual screening (VS) consist on datasets with selected known active ligands and inactive decoys for a single protein target [84]. Since information on inactive molecules is scarce in comparison to active ones, most decoys are not selected based on experimental data but are instead putative inactive compounds [85], whose selection must be made carefully so as to avoid artificial enrichment [86]. This scarcity occurs because active molecules are better described and documented, however, the opposite asymmetry is observed in nature: from a varied set of molecules which come in contact with a given protein, only a few specific ones will be active against it. Therefore, VS programs must be capable of identifying active compounds amidst a large pool of inactive ones, thus, benchmarking sets mirror this natural asymmetry by providing many putative decoys for a single known active molecule. In order to prevent bias, the active and decoys sets’ characteristics must be equally balanced: one set must not be more structurally complex or diverse than the other [87,88]; both sets should not cover small chemical spaces [84]; and there must not be any actual binders among the decoys (Latent Actives in the Decoy Set, LADS) [89]. Datasets are therefore curated in order to avoid bias as well as provide as much useful data as possible; the most widely used are described as follows.

The Directory of Useful Decoys (DUD) was created based on the principle that decoys must resemble the physical properties of the actives but be sufficiently chemically distinct to be in fact nonbinders [90]. DUD then became the gold standard benchmark for VS [91]. It was later improved into the Directory of Useful Decoys-Enhanced (DUD-E) [92], which selects decoys based on more physicochemical properties, adds more targets, and provides a tool for decoy generation based on user-input actives.

The Demanding Evaluation Kits for Objective in Silico Screening (DEKOIS) [89] was created with special attention to avoiding poorly embedded actives and LADS. A new version, DEKOIS 2.0 [93], was released two years later with additional physicochemical properties for matching decoys and an enhanced elimination of LADS.

The Maximum Unbiased Validation (MUV) [94] datasets were curated with special care for the chemical diversity of the actives set, in order to avoid over-representation of chemical entities and thus avert overestimation of performance. An exclusion of potentially unspecific active compounds was also implemented, as well as removal of actives devoid of decoys in its chemical space.

There are also databases for assessing virtual screening with specific targets: G-Protein-Coupled Receptor (GPCR) Ligand Library (GLL) and GPCR Decoy Database (GDD) [95], NRLiSt BDB for nuclear receptors [96] and MUBD-HDACs for histone deacetylases [97].

It is noteworthy that it is also possible to generate decoys for specific compounds when the target of interest is not available. User-input ligands must be provided in SMILES format, and a decoy set is curated based on their molecular properties. DecoyFinder [98] was the first application to provide this tool, searching the ZINC database for molecules similar to actives by comparing chemical descriptors. At about the same time DecoyFinder was published, DUD was upgraded to DUD-E, which also allows searching the ZINC database for decoys utilising the same search method employed to construct the database’s new target subsets. In 2017, Wang et al. [99] argued that these tools lacked computational speed for large active sets and flexible input options to avoid bias in the user-specified active set. To address these issues, they created RADER (RApid DEcoy Retriever), which selects decoys from four different databases, including ZINC.

### 2.3. Evaluation Metrics

The most widely used metrics to assess ranking performance in VS are receiver operating characteristic (ROC) curves and enrichment factors (EF). The ROC method plots the rank’s specificity and sensitivity into a curve whose area (area under the curve, AUC) ranges from 0 (worst performance) to 1 (best performance), where 0.5 reflects a randomly distributed ranking order. The calculations are made based on cut-offs throughout the whole rank, and therefore ROC reflects only overall performance [100,101]. However, when evaluating VS performance, the enrichment at the top of the rank is most important (i.e., the early recognition problem), since there can be found the molecules identified by the SF as the most probable actives [102]. EF can be used to calculate the enrichment at an early single cut-off [83] or at many cut-offs [101], which addresses the early recognition problem, however its main setback resides in the fact that its maximum value depends on the active/inactive ratio on the dataset [101,103].

It is noteworthy that by calculating Youden’s index (sensitivity + specificity − 1) for all cut-offs made in the ROC curve, one can determine the optimal threshold (i.e., the cut-off with the highest index) through which continuous binding predictions of a particular SF can be converted into to binary active/inactive classification [104].

Other metrics have been suggested and applied to better address the early recognition problem. For instance, the Robust Initial Enhancement (RIE) metric [105] applies weight to the active molecules. The active will weigh closer to 1 the better ranked it is, and its weight will fall as the rank increases. A RIE value of 1 indicates a random distribution of the rank, and its maximum value depends on the active/inactive ratio, similarly to EF. The Boltzmann-Enhanced Discrimination of Receiver Operating Characteristic (BEDROC) [102] incorporates the RIE weighing strategy into ROC curves: performance is measured in a 0 to 1 range and advantage is given to better ranked actives. One drawback of the BEDROC approach is that the magnitude of this advantage is controlled by a single parameter, which can frustrate performance comparisons between different studies [103].

No single benchmarking set or metric can be considered to be best overall for molecular docking. Rather, they are chosen differently depending on the inquiry, as well as carefully, in order to avoid biasing issues. Erroneous estimations of performance negatively impact studies and are also very hard to detect based on benchmarking results alone. Nonetheless, benchmarking datasets provide invaluable means for quality assessment of computational methods in drug discovery.

## 3. Consensus Methods

With the continued development of new scoring functions (SFs) and the improvement of well-established ones, the use of docking strategies that combine two or more SFs has become increasingly common. That is especially interesting because the various available functions perform differently across the spectrum of potential interactions, and presumably, in an ideal combination, the shortcomings of a particular function may be compensated by the others.

This strategy was first suggested by Charifson and co-workers in a study in which they benchmarked several SFs, both individually and in combination, using p38, IMPDH and HIV protease as model systems. Their approach involved taking the intersection of the top-scoring molecules according to two or three different functions available at the time and they found it provided a “dramatic reduction in the number of false positives identified by individual SFs” [106].

A consensus-docking protocol will generally differ in three major aspects: (i) the means by which the poses are obtained, (ii) the selection of the SFs, and (iii) the algorithm used to achieve the consensus. Realistically, the number of possible procedures is overwhelming, and, to date, no single protocol has been proven remarkably superior to the others. Nevertheless, it is absolutely clear that consensus methods perform consistently better when compared to individual SFs (c.f. referenced papers in Table 2).

The theoretical rationale for this was explored in 2001, soon after the first approaches, in a work in which the authors simulated an idealised computer experiment where scores were generated for a hypothetical set of 5000 compounds and the effects of consensus strategies were evaluated. The authors suggest that the improvement is largely due to the fact that the mean value of repeated samplings tends to be closer to the true value than any single sampling [107].

Although some initiatives have been explored to come up with composite scoring schemes that are applied simultaneously during the posing procedure [108], in most cases, the consensus is achieved after the conformational sampling. Moreover, it is widely accepted that the conformational sampling is not the major bottleneck in the docking process [109,110] therefore, a greater fraction of the developed methods generate the docking poses using a single algorithm and subsequently use a different set of SFs to re-assess them (Table 2 and Table 3). Nevertheless, several groups have focused on obtaining more reliable poses, for example, Ren and co-workers have explored the effects of using multiple software in the pose generation step [111]. They used a RMSD-based criterion to come up with a representative pose derived from a minimum of three and a maximum of 11 docking programs. A pose representative was selected for all possible combinations and their method achieved an increase in the success rate (pose-to-reference RMSD < 2.0 Å) of approximately 5% when compared to the best independent program.

Additionally, the concept of “consensus level” has been explored in recent works [112,113], and similarly to the previously described approach, it uses a combination of docking software to generate ligand poses, which are then clustered and the number of software that predict the same pose is taken as the consensus level. This metric can then be used to reject compounds that fail to attain a certain level and true ligands are less likely to being rejected, which, in turn increases the enrichment factors.

Another consensus posing strategy is to reject a given pose if it two or more programs fail to “converge” to that conformation. Houston and Walkinshaw have demonstrated that the success rate can be increased from ~60% to ~80% by simply rejecting a molecule if the RMSD between the poses calculated by two programs (AutoDock and VINA) is greater than 2.0 Å. The idea behind this approach is that a correct pose is more likely to be predicted by more than one algorithm, thus eliminating the misleading orientations (which could be considered false positives) [114].

Some initiatives combine consensus posing and scoring, as is the case of the VoteDock approach (and two correlated functions), proposed by Plewczynski et al., in which they combine cross-software pose conformation agreement, in the form of a voting system, with a composite scoring obtained via multivariate linear regression with results performing consistently better than individual SFs [115]. 

Besides consensus posing, many groups have focused their efforts on creating consensus scoring schemes. Very recently, Perez-Castillo and co-workers have applied the Genetic Algorithm to devise the best combination from a total of 15 SFs (or 87 scoring components) that maximises either the enrichment factor or the BEDROC value. Their results suggest that combining scoring components, instead of SFs themselves is a more effective strategy. Their algorithm, CompScore, is made available as a webserver [116].

Other reported strategies for achieving scoring function consensus are sequential docking [117,118], linear regression [119], rank-by-rank, rank-by-number, rank-by-vote [86,107,120] and standard deviation consensus [121]. Combinations of consensus docking strategies and ligand-based approaches have also been suggested [122,123]. 

Machine learning algorithms have also been employed in the determination of the consensus in recent developments. Early efforts used Random Forest algorithms to achieve consensus for 11 different SFs, outperforming the regular rank-by-rank approach in about 5%–10% and individual SFs by a far greater margin [125]. Support Vector Rank Regression (SVRR) has been suggested as a possible tool to combine seven distinct SFs (Glide- Score, EmodelScore, EnergyScore, GoldScore, ChemScore, ASPScore and PLPScore) computed using GLIDE and GOLD docking programs, and was shown to improve correct top pose prediction (RMSD < 2.0 Å) by 12.1% and correct top ligand selection by 46.3% [126]. In another study, Ericksen and collaborators used gradient boosting to derive a consensus score and benchmarked this approach using 21 targets selected from DUD-E, gradient boosting was shown to outperform traditional consensus methods (maximum, median and mean scores) and as well as the mean-variance consensus [124]. A summary of the aforementioned works can be found in Table 2.

Although molecular docking was first applied over three decades ago, it is apparent, given the virtually endless protocols, that there is still much improvement to be made in the field. In this sense, initiatives such as the Community Structure-Activity Resource (CSAR active from 2010 to 2014) [73,132] and the Drug Design Data Resource (D3R) [133,134] are invaluable as they promote the standardisation of validation datasets and metrics, as well as serve as a repository for the knowledge accumulated in the field. 

A simple comparison made with a keyword search software in the SCOPUS database for the years 1995 until 2018 (“TITLE-ABS-KEY (*software* AND docking) AND PUBYEAR > 1994 AND PUBYEAR < 2019” where the word *software* is replaced by several of the mostly employed docking programs) shows the relative prevalence of these software. Substituting the term *software* for consensus, shows that consensus methods, in spite of consistently showing superior results, are less frequently mentioned in the literature than some of the more common docking programs (at least in the searched fields, i.e., title, abstract and keywords) (Figure 2). While one could argue that this could be due to the fact that the fraction of works that indeed use consensus methods also mention other software, Figure 3, which contains the ratio of (research and conference) papers mentioning “molecular docking” OR “ligand docking” to the ones mentioning (“molecular docking” OR “ligand docking”) AND consensus, shows that the discrepancy is even more pronounced (an average of 88.36 works that cite molecular docking per each work that mentions the word consensus—Figure 3).

There is also clear disparity in the level of elaborateness between the protocols used by the groups that develop and the ones that implement these methods. As a result, the virtual screening protocols used by the latter (such as sequential docking, rank-by-number and RMSD-based pose rejection) are often less involved than the ones suggested by the former. Table 3 summarises recent works that employed consensus docking in their screening methodologies, along with the best experimentally-determined activity. Despite using more straightforward methodologies to achieve consensus, these studies show the importance of combining distinct SFs, since they have still been able to find relatively potent ligands. It appears that, easy-to-use, carefully designed and validated docking pipelines which include consensus posing and/or scoring are called for and could be widely adopted in structure-based drug design studies, both in academic and industrial settings.

## 4. Efficient Exploration of Chemical Space: Fragment-Based Approaches

### 4.1. The Chemical Space

Since it was first described in the late 1990s [135], fragment-based drug (or, less frequently, lead [136]) discovery (FBD/LD) has gained a lot of attention and many drug candidates developed with the use of such approaches have reached clinical trials [137]. The fundamental aspect that fosters its popularity is that it allows an efficient exploration of the chemical space with relatively small sampling, i.e., by combining smaller fragments that show high ligand efficiency, it is possible to design very potent ligands which would, otherwise, be dispersed in a vast pool of possible molecules. Additionally, it has been demonstrated that the probability of a given interaction between a given ligand and receptor is inversely proportional to the ligand complexity [138,139], suggesting that higher hit-rates could be achieved by screening less complex molecules.

In 2007, researchers from Reymond’s group at the University of Berne used a graph-based approach to generate all possible topologies for chemically-stable compounds presenting up to 11 atoms, and they generated a database containing near 26.4 million (2.64 × 10^7^) molecules (GDB11) [140]. Since then, they have created new sets of increasingly larger molecules, namely containing up to 13 heavy atoms (GDB13—9.7 × 10^8^ molecules) and up to 17 heavy atoms (GDB17—1.66 × 10^11^). These numbers might seem overwhelming, but not if compared to an astonishing 10^60^ estimated drug-like molecules (with up to 30 heavy atoms) [62].

Very recently, researchers from UCSF (University of California, San Francisco) have completed ultra-large campaigns, screening approximately 99 million compounds for AmpC β-lactamase and 138 million compounds for the D4 dopamine receptor, ultimately finding 30 compounds with sub-micromolar activity, including one with picomolar activity (180 pM) [141]. Endeavours of such magnitude have not been customarily undertaken, since they require great use of computational resources, therefore, fragment-based approaches can help efficiently explore the chemical space since (i) they have a small amount of degrees of freedom, leading to faster spatial sampling, (ii) they can be combined to create larger, more potent ligands, requiring reduced screening libraries to achieve comparable chemical space coverage, and (iii) the reduced complexity of fragments should lead to increased hit-rates. 

Experimentally, due to the reduced affinities, these fragments must be screened using more sensitive biophysical assays, such as Fluorescence-Based Thermal Shift, NMR Spectroscopy and Surface Plasmon Resonance [142]. Molecular docking can also be an invaluable tool for the detection of potentially interacting fragments and several examples will be discussed below. Candidate fragments detected by experimental or computational approaches are then usually evaluated through X-ray Crystallography [142] or even High Throughput X-ray Crystallography (HTX), where protein crystals are soaked in high concentrations of one or more fragments and the structure of the complex is subsequently determined [143].

### 4.2. Fragment Libraries

Some aspects must be taken into consideration when tailoring fragment libraries in order to optimise fragment-based drug design (FBDD) outcomes. For instance, because fragments are smaller, they tend to bind less tightly to the protein targets, exhibiting lower potency values. Therefore, it is advantageous to use size-normalised parameters, such as Ligand Efficiency (LE) [144], Binding Efficiency Index (BEI) [145] or Fit Quality (FQ) [146], to prioritise the evaluated molecules. These can then serve as objective parameters to a successful subsequent lead optimisation [147]. Secondly, Harren Jhoti’s group has suggested an adjusted set of rules [148] (or guidelines [149]), termed Rule of Three (RO3), derived from hits obtained via High Throughput X-ray Crystallography (HTX) and inspired by Lipinski’s Rule of Five [150]. These stemmed from the observation that successful hits customarily present molecular weight under 300 Da, three or fewer hydrogen bond donors, three or fewer hydrogen bond acceptors, clogP under three and, additionally, three or fewer rotatable bonds and a polar surface area under 60 Å^2^. These guidelines can help filtering fragment libraries for efficient screening, both experimentally and computationally. A third matter worth noting is the reported “lack of tri-dimensionality” in fragment libraries, which can hinder the development of ligands with high affinity for certain classes of targets [151].

Fragment libraries can be generic or generated ad hoc (targeted, or focused libraries). Many of the generic libraries are commercially available on demand, and thus may be readily used in experimental screens, and the compound chemical structures are usually also available as Structure-Data Files (SDF), which can be straightforwardly converted to other structural formats, such as MOL2, PDB and PDBQT and used for virtual screening (cf. Verheij [152] work on lead-likeness for sources of such libraries). Fragment libraries usually contain 10^2^ to 10^4^ molecules, which are generally compliant with the RO3 and are idealised to maximise attributes such as solubility, chemical stability, scaffold complexity, tri-dimensionality and tractability [151,153,154]. Tractability-guided fragmentation algorithms and pipelines can be used to generate specialised fragment libraries starting from collections such as the World Drug Index (which has been fragmented using the RECAP algorithm [155]) or natural products libraries [156].

The combination of fragments into a larger molecule has been classified into four distinct categories, namely Merging, Linking, Growing and “SAR by catalogue” [153]. In fragment merging, two fragments occupying an overlapping site are joined together to obtain a larger molecule with higher affinity. Conversely, in fragment linking, the fragments are usually bound to two distinct binding pockets (or sub-pockets) and are joined together via the construction of a linker fragment, that ideally allows the maintenance of the initial orientation of the fragments. Fragment growing consists of the design and incorporation of new functional groups that are expected to form new interactions with the receptor, thus increasing the binding affinity. Finally, the “SAR by catalogue” is particularly interesting from the virtual screening angle due to its simplicity; in this approach, a fragment initially detected (and ideally confirmed by experimental techniques) is then used as an “anchor” to query a database for larger molecules that contain the original fragment. Thus, effectively, this strategy is largely used to create more focused libraries.

### 4.3. Molecular Docking in FBDD

Many groups have used FBDD to idealise potent ligands for disease-modifying protein targets with extensive use of molecular docking and virtual screening approaches. In a study developed by Chen and Shoichet, a fragment-based approach was used as an alternative to a lead-like virtual screen campaign, obtaining increased hit rates for β-Lactamase inhibitors, and ultimately yielding hits in the low µM range [157]. This indicates that even using similar docking protocols, fragment-based approaches can yield more accurate initial hits when compared to lead-like molecules screening.

These computationally-driven works reflect some of the experimental strategies discussed above, since the initial screens for promising fragments are usually followed by a fragment-joining step, which can be accomplished in a manual [158] or automated [159] way. Recently, Park et al. have been able to design nanomolar-range inhibitors for the protein Glycogen Synthase Kinase-3 β, using AutoDock [32] as the initial tool to perform virtual screening of fragment libraries in three independent subsites and LigBuilder [160] as the tool to connect a series of selected fragments [159].

Employing the “SAR by catalogue” method, Zhao and co-workers, after initial filtering of the ZINC database, have used an in-house docking solution to prioritise anchor fragments that bind the BRD4 bromodomain, which were then used to further interrogate the database and retrieve compounds containing the selected moieties, ultimately finding compounds with activity in the low micromolar range (7.0–7.5 µM) [161]. Using a similar “anchor-based” analogue search approach, Rudling and co-workers have used Dock3.6 to find inhibitors in the low micromolar range for MTH1 protein, an interesting cancer target, and in a second round of prospection for commercially available analogues, they managed to further optimise the initial hits to achieve IC_50_ values as low as 9 nM [162].

Hernandez et al. have suggested non-nucleoside inhibitors of flaviviral methyltransferases (Zika virus and Dengue virus NS5MTase) presenting IC_50_ ~20 µM by screening a focused library constructed using a knowingly binding core substructure, encoded organic chemistry rules and commercially available building blocks. The authors refer to this approach as fragment-growing [163].

The successful combination of fragment-based virtual screening and NMR screening has also been reported. Fjellström et al. have identified Activated Factor XI inhibitors using Glide to prioritise 1800 molecules (out of 6.5 × 10^3^ from AstraZeneca screening collection with molecular weight (MW) < 250 g/mol) for NMR fragment screening. Subsequent structure-based expansion and re-scoring of 13 NMR hits yielded a compound with activity of 1.0 nM [158]. Using an inverted approach, Akabayov and co-workers used an initial NMR screen of a library containing 1000 fragments to identify moieties that bind T7 DNA primase, the two most promising hits were then used to query ZINC database, once more reflecting the “SAR by catalogue” approach, and the selected molecules (approximately 3000 per scaffold) were docked to DNA primase structure, using Autodock4. About half of the 16 selected compounds showed inhibitory activities [164]. 

Amaning and co-workers prospected for MEK1 inhibitors carrying out a virtual screening campaign of approximately 10^4^ molecules, used to prioritise fragments to be further characterised by differential scanning fluorimetry (DSF), surface plasmon resonance and X-ray crystallography. Interestingly, a parallel biochemical screening of the same library showed that the 5% of the best scoring molecules in the virtual screening contained 30% of the biochemical hits and, according to the authors, this indicates that the VS–DSF combination can used to ‘jump-start’ a project in an early phase when a biochemical or other biophysical assays are not available [165]. Additionally, it has been suggested that characteristics such as novelty and potency are likely to differ considerably between hits determined by experimental screening and those determined by virtual screening [166].

Besides prospecting for new molecules in fragment-based VS campaigns, molecular docking is extensively used to hypothesise interaction modes and better characterise the ligand-receptor interactions [167,168,169], and remains an invaluable asset in the drug development toolkit.

## 5. Machine Learning-Based Approaches

Scoring and ranking candidate molecules through binding affinity prediction is the most challenging aspect of molecular docking and VS. Classical SFs must simplify and generalise many aspects of the receptor-ligand interaction in order to maintain efficiency, approachability and accessibility [27]. Moreover, these SFs employ linear regression models: parametric supervised learning methods, which assume a specific predetermined functional form [170]. In other words, parametric methods fit the input variables (such as van der Waals and electrostatic energy terms) to the output (binding energy score) into a function whose form is already specified, and which is adjusted during the development of the SF in a theory-inspired fashion [77]. This rigid scheme often results in unadaptable SFs which fail to capture intrinsic nonlinearities in the data and therefore underperform in situations not accounted for in their formulation [77,171].

Alternatively, nonparametric machine learning (ML) algorithms (often referred to as just “machine learning”) can be used to replace [77,172,173,174] or improve [82,175,176,177,178] predetermined functional forms in classical SFs for binding affinity predictions. They have also been successfully applied in binders/nonbinders identification in virtual screening [175,179,180,181] and native pose prediction [126,172,182].

ML methods are divided into two broad groups: supervised and unsupervised learning. Unsupervised learning algorithms are employed to model the training data when there is no output available. Thus, these algorithms are commonly used for clustering data based on the degree of similarity between their features, for detecting associations between the data points, and for density estimations. In supervised learning, however, the output variables are known and provided to the algorithm along with the input for training. In nonparametric supervised learning, no functional form is assumed. It is then possible to infer the correlations between input and output from the training data itself and utilise it to predict the output for datasets of which the outcomes are unknown [170].

This allows for more diverse and accurate SFs: more features from the docked complex can be accounted for implicitly, therefore skirting modelling assumptions and necessary generalisations of classical SFs [77,82,171]. Moreover, the adeptness of the ML algorithm can be adjusted by tailoring the training dataset. For instance, increasing the diversity of the training complexes results in ML SFs with greater comprehensiveness. In fact, it has been shown that increasing the size of the training set boosts the scoring function’s performance [82,172,183].

This contrasts greatly with classical SFs, whose parametric behaviour remains unable to improve performance with larger training datasets [82]. On the other hand, increasing the level of feature detail in training sets comprising of similar complexes may provide greater discrimination power when studying such data [183,184].

### 5.1. Protein Target Types: Generic and Family-Specific

Machine learning SFs can be considered family-specific or generic. It has been shown that family-specific SFs can outperform most accurate generic ones at said protein family’s predictions [183,184]. Until recently, however, it was not clear whether a family-specific SF carried any advantages over generic ones whose training includes all complexes and features utilised in training the former [83]. It was later shown that random forest trained with family-specific data only slightly outperformed the universal model. This outperformance grew, however, when predicting more difficult targets with less active ligands [185]. In a 2018 study with deep learning neural networks, Imrie et al. [183] showed that family-specific models trained with a subset of the entire dataset outperformed universally trained models, and that only limited family data was required for this outperformance to occur. For each different protein family, the importance of the features used to describe the data varies [184], therefore, specific SFs are able to better assimilate these characteristics as a result of dealing with less broad and more nuanced data [183,184,185].

Machine learning SFs have been regarded both as knowledge-based [186,187] and empirical [188]. However, it is important to note that this categorisation has extensively been used in regard to classical SFs, and therefore it should not obscure the fact that there is a more fundamental difference between them: the former consists of nonparametric and the latter of parametric learning (Figure 4).

### 5.2. Experiment Types: Binding Affinity Prediction and Virtual Screening

SFs designed for binding affinity predictions can also be used for virtual screening experiments, as long as the predicted results are ordered from best to worst binding score. If a binary active/inactive distinction is desired, one can establish an optimal activity threshold score by analysing the SF’s performance on a benchmarking dataset (c.f. Benchmark Datasets section). However, ML classifiers built for VS may present better discrimination since their training utilises datasets specific for portraying virtual screening circumstances i.e., they are often trained on data derived from in silico approaches (as opposed to crystal structures of complexes) which do not always represent the correct binding mode, and the features from docked decoy molecules are also used for training [189].

### 5.3. Algorithms and Feature Selection

Feature selection plays an important role in the development of ML methods. Selecting a subset of features which are appropriate and effective for characterising the data not only improves prediction performance, but also reduces computational expense and facilitates the understanding of the intrinsic patterns underlying the data [190].

The first ML SF to outperform classical SFs [83], random forest (RF)-Score [77], utilised the random forest (RF) algorithm with intermolecular interaction features comprised of the number of a particular protein-ligand atom type pair interacting within a certain distance range [77]. Other descriptors such as energy terms from classical SFs, solvent accessible surface area, entropy, hydrophobic interactions and chemical descriptors have been applied by works such as those of Springer et al. (PostDOCK) [181], Pereira et al. (DeepVS) [177], Jiménez et al. (Kdeep) [78], Durrant et al. (NNScore) [79], Koppisetty et al. [191] and Liu et al. (B2BScore) [192] with various degrees of success. It has been shown that richer and more precise chemical descriptors do not generally result in more accurate predictions [193], and that different SFs have very different responses to an increase in the number of features [171].

Other ways of describing the data have been explored. For instance, Kundu et al. [194] utilised fundamental molecular descriptors for the proteins and the ligands, without any intermolecular interaction features, which circumvents the need for binding pose information. Srinivas et al. [195] utilised collaborative filtering, an algorithm extensively employed for recommendation systems (i.e., predicting appropriate online costumer recommendations), to bypass the explicit definition of receptor and ligand features. The similarities in the data are inferred only based on the results of the recorded binding assays.

### 5.4. Deep Learning

Deep learning neural networks have recently been applied to pose prediction and ranking [78,173,177,183,196]. Convolutional neural networks, which are known to present outstanding image recognition capabilities [197], in molecular docking, have been explored mainly by featurising the protein-ligand complexes as three-dimensional grids. Deep learning SFs have yielded state-of-the-art results [78,183,196], comparable to and even surpassing those achieved by random forest, support vector machines, and boosted regression trees, the other non-neural network algorithms reported to be the most accurate for protein-ligand scoring [171,198].

### 5.5. Recent Applications and Perspectives

It is noteworthy that although the current ML techniques already promise to advance computational drug discovery, some limitations still need to be addressed. For instance, larger amounts of data are still required to reach optimal deep learning performance, and it is not clear whether at some point learning saturation can occur [183]. Furthermore, complex nonparametric learning models can be difficult to interpret. Sieg et al. [199] very recently pointed out that bias is being implicitly learned from standard benchmarking sets, and suggested guidelines to avoid fallacious models.

ML SFs for molecular docking have only recently been introduced. Naturally, most studies are dedicated to assessing and improving their predictive powers, and not as many have applied them in drug discovery and repurposing experiments. Nonetheless, existing prospective studies show positive results (Table 4). In 2011, Kinnings et al. [175] created a support vector machine-based SF to improve binding affinity prediction from classical SFs and used it to identify that phosphodiesterase inhibitors could potentially be repurposed towards *Mycobacterium tuberculosis* protein InhA. One year later, Zhan et al. [123] used support vector machine to integrate classical docking scores, interaction profiles and molecular descriptors to identify six novel Akt1 inhibitors. Durrant et al. (2015) used NNScore, a neural network SF, to describe 39 novel oestrogen-receptor ligands, whose activities were experimentally confirmed [200].

Among the ML SFs mentioned in this section, those readily accessible for use are the following: RF-Score; NN-Score; Ragoza et al.’s final optimised model architecture; DLScore; and kDEEP. These are available as downloadable standalone programs, with the exception of Kdeep, which can be found at playmolecule.org. If online docking is desired, CSM-lig [201] (for binding affinity predictions) is also available as a web-server. To the best of our knowledge, none of these SFs have been integrated into docking programs such as the ones summarised in Table 4.

Machine learning methods have shown positive results, as well as promising room for more enhancement. In addition, the availability of benchmarking data for training and testing is likely to be further expanded, which will consequently improve the predictive power of these techniques. Therefore, nonparametric machine learning is potentially the next step to drastically improve molecular docking predictiveness and accuracy.

## 6. Conclusions

Molecular docking has been established as a pivotal technique among the computational tools for structure-based drug discovery. Here we addressed key aspects of the methodology and discussed recent trends in the literature for advancing and employing the technique for successful drug design. Benchmarking sets and the various metrics available are crucial for validating performance gains achieved by new docking software but must be carefully chosen since no single one can be regarded as the absolute best for molecular docking. A significant improvement in the performance of all docking software can be achieved by employing multiple SFs for consensus posing and/or scoring. As reviewed here, there is a plethora of protocols for consensus docking to be explored by the user.

FBDD emerged as a successful paradigm for developing new drugs, combining the serendipity of target-based high throughput screening with the rationality of structure-based drug design approaches. Molecular docking has important roles in FBDD, from planning and prioritisation of fragment library composition to finding analogues with improved binding affinities through large-scale VS of compound libraries.

ML is a branch of artificial intelligence that has gained much attention in diverse fields of science and technology and molecular docking methods are also taking advantage of this pulsating area. Although recent, the flexibility of ML in modelling data has already rendered more diverse and accurate SFs implicitly accounting for more features from the docked complex.

## Figures and Tables

**Figure 1 ijms-20-04574-f001:**
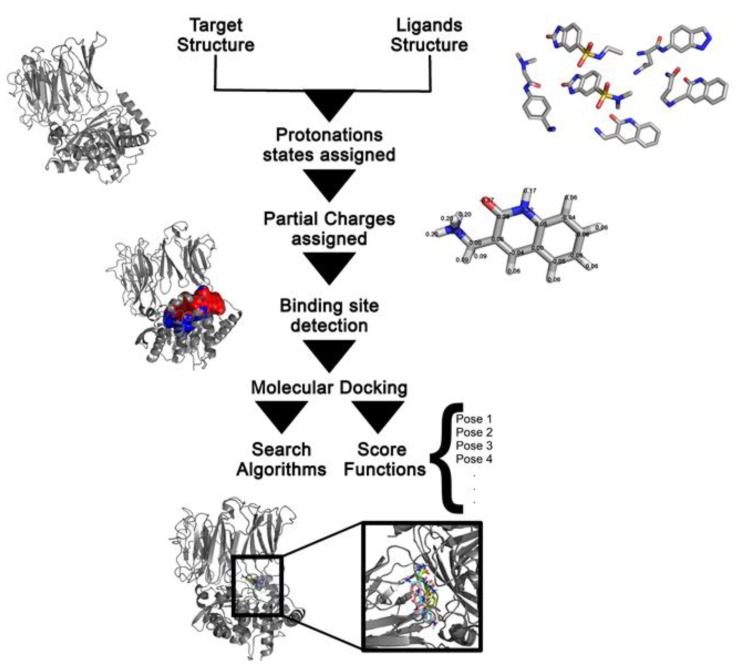
General workflow of molecular docking calculations. The approaches normally start by obtaining 3D structures of target and ligands. Then, protonation states and partial charges are assigned. If not previously known, the target binding site is detected, or a blind docking simulation may be performed. Molecular docking calculations are carried out in two main steps: posing and scoring, thus generating a ranked list of possible complexes between target and ligands.

**Figure 2 ijms-20-04574-f002:**
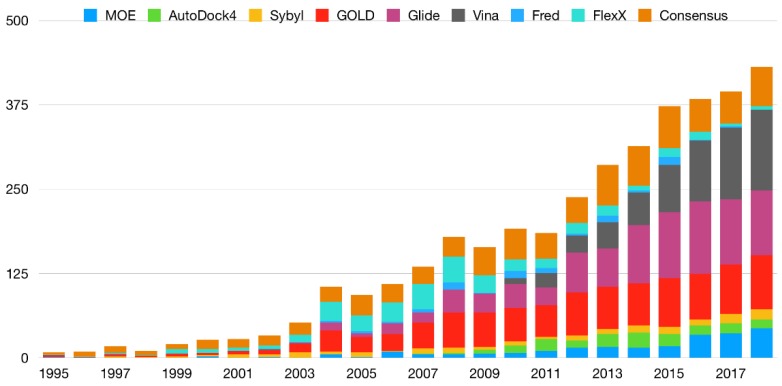
Scopus search results for the query “TITLE-ABS-KEY (*software* AND docking) AND PUBYEAR > 1994 AND PUBYEAR < 2019” where the word software is substituted for one of the eight most common docking software or by the word consensus.

**Figure 3 ijms-20-04574-f003:**
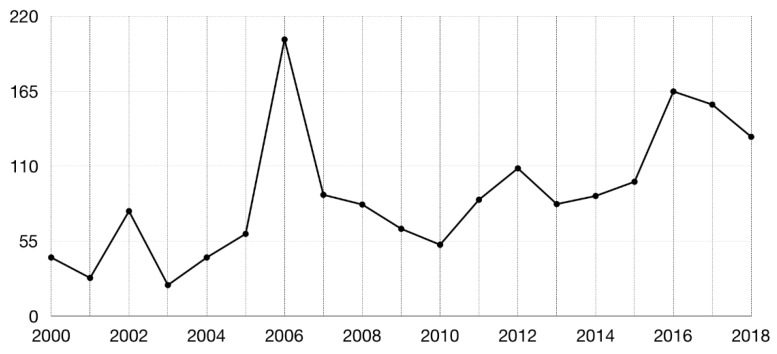
Ratio of the numbers of papers containing either the expression “molecular docking” or “ligand docking” to the number of papers containing either of the two expressions AND the word consensus.

**Figure 4 ijms-20-04574-f004:**
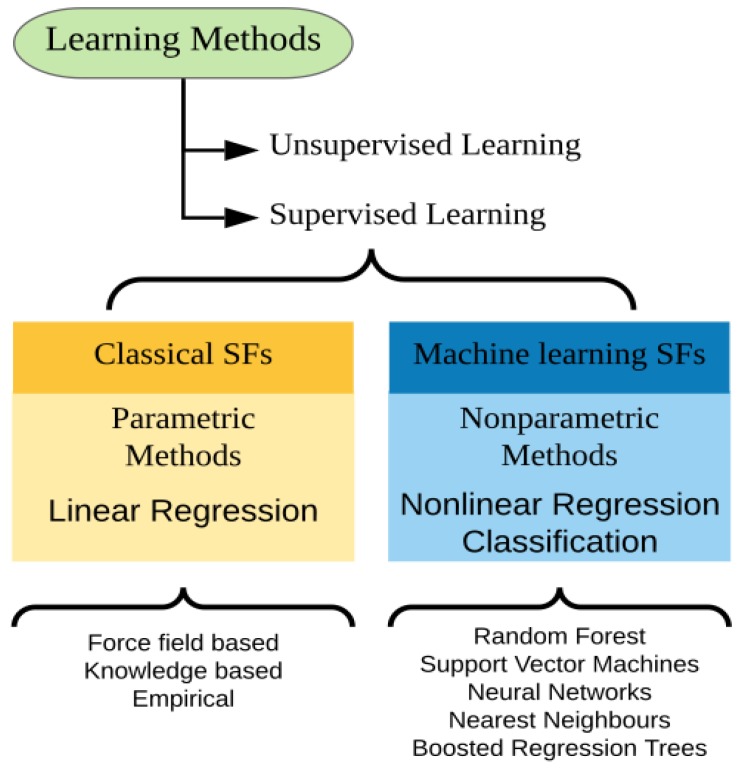
Learning methods can be broadly divided into supervised learning, when there is data available for training and parameterisation; and unsupervised learning, when there is no such data. Unsupervised learning cannot be used for binding affinity predictions and virtual screening. Supervised learning, on the other hand, can be divided into parametric and nonparametric learning. Parametric learning assumes a predetermined functional form, as observed in linear regression, and is the method employed in classical scoring functions. Nonparametric learning, or just machine learning, does not presume a predetermined functional form, which is instead inferred from the data itself. It can yield continuous output, as in nonlinear regression, or discrete output, for classification problems such as binders/nonbinders identification.

**Table 1 ijms-20-04574-t001:** Molecular docking software.

Software	Posing	Scoring	Availability	Reference
Vina	Iterated Local Search + BFGS Local Optimiser	Empirical/Knowledge-Based	Free (Apache License)	Trott, 2010 [3]
AutoDock4	Lamarckian Genetic Algorithm, Genetic Algorithm or Simulated Annealing	Semiempirical	Free (GNU License)	Morris, 2009; Huey, 2007 [31,32]
Molegro/MolDock	Differential Evolution (Alternatively Simplex Evolution and Iterated Simplex)	Semiempirical	Commercial	Thomsen, 2006 [9]
Smina	Monte Carlo stochastic sampling + local optimisation	Empirical (customisable)	Free (GNU License)	Koes, 2013 [33]
Plants	Ant Colony Optimisation	Empirical	Academic License	Korb, 2007; Korb, 2009 [34,35]
ICM	Biased Probability Monte Carlo + Local Optimisation	Physics-Based	Commercial	Abagyan, 1993; Abagyan, 1994 [36,37]
Glide	Systematic search + Optimisation (XP mode also uses anchor-and-grow)	Empirical	Commercial	Friesner, 2004 [38]
Surflex	Fragmentation and alignment to idealised molecule (Protomol) + BFGS optimisation	Empirical	Commercial	Jain, 2003; Jain 2007 [39,40]
GOLD	Genetic Algorithm	Physics-based (GoldScore), Empirical (ChemScore, ChemPLP) and Knowledge-based (ASP)	Commercial	Jones, 1997; Verdonk 2003 [6,7]
GEMDOCK	Generic Evolutionary Algorithm	Empirical (includes pharmacophore potential)	Free (for non-commercial research)	Yang, 2004 [41]
Dock6	Anchor-and-grow incremental construction	Physics-based (several other options)	Academic License	Allen, 2015 [42]
GAsDock	Entropy-based multi-population genetic algorithm	Physics-based	*	Li, 2004 [43]
FlexX	Fragment-Based Pattern-recognition (Pose Clustering) + Incremental Growth	Empirical	Commercial	Rarey, 1996; Rarey, 1996b [8,44]
Fred	Conformer generation + Systematic rigid body search	Empirical (defaults to Chemgauss3)	Commercial	McGann, 2011 [45]
DockThor	Steady-state genetic algorithm (with Dynamic Modified Restricted Tournament Selection method)	Physics-based + Empirical	Free (Webserver)	De Magalhães, 2014 [4,25]

* Availability is unclear.

**Table 2 ijms-20-04574-t002:** Consensus docking methods.

Source	T ^a^	Posing ^b^	F ^c^	Consensus Strategy	Analysis	Ref.
**DUD-E/** **PDB**	102/3	4	4	Standard Deviation Consensus (SDC), Variable SDC (vSDC)	Rank/Score curves Hit recovery count	Chaput, 2016 [121]
**DUD-E**	21	8	8	Gradient Boosting	EF, ROCAUC	Ericksen, 2017 [124]
**PDBBind** **DUD**	228/1	Vina, AutoDock	2	Compound rejection if pose RMSD > 2.0 Å	Success rate	Houston, 2013 [114]
**PDB**	3	GAsDock	2	Multi-Objective Scoring Function Optimisation	EF	Kang, 2019 [108]
**mTOR ^d^ Inhibitors**	1	Glide	26	Linear Combination	BEI Correlation	Li, 2018 [119]
**PDB**	220	FlexX	9	Several ^e^	Compression and Accuracy	Oda, 2006 [120]
**DUD-E**	102	Dock 3.6	15	Genetic Algorithm used to combine SF components	EF, BEDROC	Perez-Castillo, 2019 [116]
**PDBBind**	1300	7	7	RMSD-based pose consensus, multivariate linear regression	Success rate	Plewczynski, 2011 [115]
**DUD**	35	10	10	Compound rejection based on RMSD consensus level	EF	Poli, 2016 [112]
**PDBBind**	3535	11	11	Selection of representative pose with minimum RMSD	Success rate	Ren, 2018 [111]
**PDB**	100	AutoDock	11	Supervised Learning (Random Forests), Rank-by-rank	Average RMSD, Success rate	Teramoto, 2007 [125]
**PDB** **DUD**	130/3	10	10	Compound rejection based on RMSD consensus level	EF, ROCAUC	Tuccinardi (2014) [113]
**PDBBind CSAR**	421	Glide	7	Support Vector Rank Regression	Top pose /Top Rank	Wang, 2013 [126]
**PDB**	4	GEMDOCK GOLD	2	Rank-by-rank, Rank-by-score	Rank/Score curve, GH Score, CS index	Yang, 2005 [127]

^a^ Total number of targets used in the assay; ^b^ Posing software used. If more than two software were used, than only the number is indicated; ^c^ Number of scoring functions used; ^d^ In this study, the dataset was composed of 25 mammalian target of rapamycin (mTOR) kinase inhibitors retrieved from the literature and six mTOR crystal structures retrieved from PDB; ^e^ The purpose of this study was to evaluate several different consensus strategies (e.g., rank-by-vote, rank-by-number, etc).

**Table 3 ijms-20-04574-t003:** Recent works using consensus docking approaches.

Target	Lig.	Posing	F ^a^	Consensus Strategy	Hits/Test	Best Activity (IC_50_)	Ref.
**EBOV Glycoprotein**	3.57 × 10^7^	VINA, FlexX	2	Sequential Docking	-	-	Onawole, 2018 [117]
**β-secretase (BACE1)**	1.13 × 10^5^	Surflex	12	Z-scaled rank-by-number Principal Component Analysis	2/20	51.6 μM	Liu, 2012 [128]
**c-Met Kinase**	738	2	2	Sequential Docking Compound rejection if pose RMSD > 2.0 Å	-	-	Aliebrahimi, 2017 [118]
**Acetylcholinesterase**	14,758	4	4	vSDC [121]	12/14	47.3 nM	Mokrani, 2019 [129]
**PIN1**	32,500	10	10	Compound rejection based on RMSD consensus level	1/10	13.4 μM 53.9 µM ^c^	Spena, 2019 [130]
**Akt1**	47	LigandFit	5	Support Vector Regression	6/6 ^b^	7.7 nM	Zhan, 2014 [123]
**Monoacylglycerol Lipase (MAGL)**	4.80 × 10^5^	4	4	Compound rejection based on RMSD consensus level	1/3	6.1 µM	Mouawad, 2019 [131]

^a^ Number of scoring functions used; ^b^ This work consisted of a Quantitative Structure-Activity Relationship (QSAR) model using consensus docking as descriptors. Six compounds were designed, synthesised and tested, exhibiting IC_50_ values between 7.7 nM and 4.3 μM; ^c^ First IC_50_ value: inhibitory activity against PIN1 isomerisation. Second IC_50_ value: inhibitory effects on ovarian cancer cell lines.

**Table 4 ijms-20-04574-t004:** Recent developments using machine learning (ML) algorithms in molecular docking.

SF Name	ML Algorithm	Training Database	Best Performance	Generic or Family Specific	Type of Docking Study	Reference
RF-Score	RF ^a^	PDBbind	Rp ^b^ = 0.776	Generic	BAP ^c^	Ballester 2010 [77]
B2BScore	RF	PDBbind	Rp = 0.746	Generic	BAP	Liu 2013 [192]
SFCScore^RF^	RF	PDBbind	Rp = 0.779	Generic	BAP	Zilian, 2013 [202]
PostDOCK	RF	Constructed from PDB	92% accuracy	Generic	VS ^d^	Springer, 2005 [181]
-	SVM ^e^	DUD	-	Both	VS	Kinnings, 2011 [175]
ID-Score	SVR ^f^	PDBbind	Rp = 0.85	Generic	BAP	Li, 2013 [203]
NNScore	NN ^g^	PDB; MOAD; PDBbind-CN	EF = 10.3	Generic	VS	Durrant, 2010 [79]
CScore	NN	PDBbind	Rp = 0.7668 (gen.) Rp = 0.8237 (fam. spec.)	Both	BAP	Ouyang, 2011 [174]
-	Deep NN	CSAR, DUD-E	ROCAUC = 0.868	Generic	VS	Ragoza, 2017 [196]
-	Deep NN	DUD-E	ROCAUC = 0.92	Both	VS	Imrie, 2018 [183]
DLScore	Deep NN	PDBbind	Rp = 0.82	Generic	BAP	Hassan, 2018 [173]
DeepVS	Deep NN	DUD	ROCAUC = 0.81	Generic	VS	Pereira, 2016 [177]
Kdeep	Deep NN	PDBbind	Rp = 0.82	Generic	BAP	Jiménez, 2018 [78]

^a^ Random Forest; ^b^ Pearson’s Correlation Coefficient; ^c^ Binding Affinity Prediction; ^d^ Virtual Screening; ^e^ Support Vector Machine; ^f^ Support Vector Regression; ^g^ Neural Network.

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
