# Peer review of "Key Topics in Molecular Docking for Drug Design"

_ijms, 2019, doi:10.3390/ijms20184574_

Round 1

Reviewer 1 Report

The review from Torres et al., deals with one of the most used methodologies for structure-based drug discovery (i.e., protein-ligand docking). The review is interesting and touches some of the important parts of the methodology but, in my opinion, could be better structured.

Moreover, I require some changes in the manuscript before accepting its publication:

The authors refer to "configuration" in some parts of the text (e.g., line 35). Configuration refers to the relative location of substituent around a chiral atom. It is not possible to change from one configuration to another one (e.g. from R to S) without breaking and forming covalent bonds. Then, the authors have to replace "configuration" with "conformation" at the text.

Line 69: "equal length cubes" is not adequate because the grid that encompasses a binding site does not need to be a cube (usually needs to be a parallelepiped).

Line 75: "the 3D atomic coordinates of these compounds can be predicted from the 2D structures" is not correct. The coordinates can be obtained from the 2D structures but not predicted.

When looking for the poses of a ligand on the binding site, there are some docking programs like eHiTS (J Mol Graph Model. 2007 Jul;26(1):198-212; Curr Protein Pept Sci. 2006 Oct;7(5):421-35) that divided the ligand into their rigid parts, dock them and then reconnect with the flexible parts not previously docked. This approach needs to be considered also in the review.

I would like to see a Table that describes the features of the most popular protein-ligand docking programs/servers (e.g., type of scoring function, availability, etc.).

It is very strange to me that in the review there is no mention of the different strategies to deal with binding site flexibility (e.g., ensemble docking, etc.)

It is well documented that coupling a structure-based pharmacophore to a protein-ligand docking can significantly improve pose prediction. The authors need to discuss this and include some examples of the literature of this coupling.

The authors need to include also a mention of the use of constraints during protein-ligand docking, how these constraints are obtained and how they can influence the results.

DUD, DUD-E and DEKOIS provide decoys for a limited number of targets. Then, the authors need to mention those programs/servers that can provide a set of decoys when the input is a set of ligands (e.g. actives for the same target)

A more thorough description of the decoys databases is absolutely necessary. Specially mentioning the ratio active/decoys and why this ratio is strongly biased towards the decoys.

Which protein-ligand docking programs/servers use (or can use) ML scoring functions? I have not found this information in the manuscript ...

Author Response

The authors refer to "configuration" in some parts of the text (e.g., line 35). Configuration refers to the relative location of substituent around a chiral atom. It is not possible to change from one configuration to another one (e.g. from R to S) without breaking and forming covalent bonds. Then, the authors have to replace "configuration" with "conformation" at the text.

We understand the reviewer concern but as a matter of clarification, in statistical thermodynamics “configuration” refers to the number of particle positions in the system. For a molecule, it may represent its number of conformations. Early docking programs such as UCSF Dock adopted the term “configuration” to denote the orientation and conformation of a ligand. Hence, we believe the term “configuration” is appropriate in this context. Nevertheless, the same confusion made by the reviewer can be done by other readers. Therefore, we decided to solely adopt in this review the term “pose”, another term from the literature to denote the solutions of a docking program. Note that a “pose” is a combination of orientation and conformation of a ligand in a binding site. Hence, using “conformation” as a replacement for “configuration” would also not be appropriate.

Line 69: "equal length cubes" is not adequate because the grid that encompasses a binding site does not need to be a cube (usually needs to be a parallelepiped).

We changed the sentence to: “This approach speeds up the docking runs and basically consists of the discretization of the binding site”.

Line 75: "the 3D atomic coordinates of these compounds can be predicted from the 2D structures" is not correct. The coordinates can be obtained from the 2D structures but not predicted.                    

We changed the sentence to: “the 3D atomic coordinates of these compounds can be obtained from the 2D structures”.

When looking for the poses of a ligand on the binding site, there are some docking programs like eHiTS (J Mol Graph Model. 2007 Jul;26(1):198-212; Curr Protein Pept Sci. 2006 Oct;7(5):421-35) that divided the ligand into their rigid parts, dock them and then reconnect with the flexible parts not previously docked. This approach needs to be considered also in the review.                              

In the introduction section, we have mentioned the eHits software (which is now also displayed in Table 1) and altered the text to: "Systematic search algorithms explore each ligand’s degree of freedom. As the number of free rotatable bonds increases, the number of evaluations can undergo a combinatorial explosion [16,26,27]. This class of search algorithms can be subdivided in exhaustive, incremental construction (which relies on the fragmentation of the ligand) and conformational ensemble [26]⁠. FlexX [8] and eHits [28], for example, employ fragment-based approaches with systematic algorithms (incremental construction and graph matching, respectively)."

I would like to see a Table that describes the features of the most popular protein-ligand docking programs/servers (e.g., type of scoring function, availability, etc.).

We thank the referee for this suggestion, and we appreciate the value it has added to the work overall. We have added such a table in the introductory section trying to summarize the currently used software comprehensively. Particularly, all software mentioned in the review have been added.

It is very strange to me that in the review there is no mention of the different strategies to deal with binding site flexibility (e.g., ensemble docking, etc.)

Unfortunately, a comprehensive section on the topic would cause the manuscript to surpass the word limit imposed by the journal, but we added the following paragraph to briefly address this issue in section 1.1: "Another widely known shortcoming of traditional docking methodologies is the poor modelling of receptor flexibility [61–63] . Some docking algorithms are able to partially mitigate this issue by allowing side-chain movement of active-site residues. Nevertheless, larger conformational changes might be triggered upon ligand-binding or might be a prerequisite to the binding event itself. A strategy, usually referred to as Receptor Ensemble Docking (or simply Ensemble Docking) is the most frequently used to model those scenarios. It is based on the concept of Conformational Selection and consists in using multiple conformations of the receptor molecule, that can be obtained via different methods, such as MD simulations [64,65], Normal Mode Analysis [66], and even by using alternative experimentally-determined receptor conformations[67]. It is worth noting that some software, such as GOLD and Glide have implemented functionality to execute this type of analysis."

It is well documented that coupling a structure-based pharmacophore to a protein-ligand docking can significantly improve pose prediction. The authors need to discuss this and include some examples of the literature of this coupling.

We added the paragraph: "Algorithms were also developed to use information from protein and ligand pharmacophores. Those algorithms try to match the distances between each of the ligand's and protein's pharmacophoric points [29]. The software FLEXX-PHARM is an extended version of FLEXX and applies pharmacophoric features as constraints into a docking calculation [30]."

The authors need to include also a mention of the use of constraints during protein-ligand docking, how these constraints are obtained and how they can influence the results.

In the introduction section, we have expanded the paragraph that mentions binding-site-detecting algorithms and added an extra paragraph mentioning distance restraints. "Several available software can be used to detect binding sites. MolDock [9], for example, uses an integrated cavity detection algorithm to identify potential binding sites. DoGSiteScorer is an algorithm that determines possible pockets and their druggability scores, which describe the potential of the binding site to interact with a small drug-like molecule [13]⁠. Fragment Hotspot Maps [14] uses small molecular probes to identify surface regions in the receptor that are prone to interact with small molecules. These predicted interaction sites can then be provided as the center of the sampling space.

Moreover, information derived from such hotspots or even from previous experimental knowledge (e.g. NMR, mass spectrometry) can be used to generate distance restraints, which is known to greatly increase protein-small molecule docking accuracy [15]."

DUD, DUD-E and DEKOIS provide decoys for a limited number of targets. Then, the authors need to mention those programs/servers that can provide a set of decoys when the input is a set of ligands (e.g. actives for the same target)

According to the reviewer's request, we have included the following paragraph at the end of topic 2.2: “It is noteworthy that it is also possible to generate decoys for specific compounds when the target of interest is not available. User-input ligands must be provided in SMILES format, and a decoy set is curated based on their molecular properties. DecoyFinder [98] was the first application to provide this tool, searching the ZINC database for molecules similar to actives by comparing chemical descriptors. At about the same time DecoyFinder was published, DUD was upgraded to DUD-E, which also allows searching the ZINC database for decoys utilising the same search method employed to construct the database's new target subsets. In 2017, Wang et al. [99] argued that these tools lacked computational speed for large active sets and flexible input options to avoid bias in the user-specified active set. To address these issues, they created RADER (RApid DEcoy Retriever), which selects decoys from four different databases, including ZINC.”

A more thorough description of the decoys databases is absolutely necessary. Specially mentioning the ratio active/decoys and why this ratio is strongly biased towards the decoys.

In order to incorporate the matter pointed out by the referee, the first paragraph in topic 2.2 was expanded: "This scarcity occurs because actives are better described and documented, however, the opposite asymmetry is observed in nature: from a varied set of molecules which come in contact with a given protein, only a few specific ones will be active against it. Therefore, VS programs must be capable of identifying actives among many more inactives, thus, benchmarking sets mirror this natural asymmetry by providing many putative decoys for a single known active molecule."

Which protein-ligand docking programs/servers use (or can use) ML scoring functions? I have not found this information in the manuscript …

To clarify this matter we have added the following paragraph as second to last in topic 5.5: “Among the ML SFs mentioned in this section, those freely accessible for use are the following: RF-Score; NN-Score; Ragoza et al.'s final optimized model architecture; DLScore; and kDEEP. These are available as downloadable standalone programs, with the exception of Kdeep, which can be found at  playmolecule.org. If online docking is desired, CSM-lig [Pires2016] (for binding affinity predictions) is also available as a web-server. To the best of our knowledge, none of these SFs have been integrated into docking programs such as the ones summarised in table 4.”

Reviewer 2 Report

The review article on key topics concerning molecular docking in drug design by Torres et al. is well-organized and comprehensive. Of course such as broad theme cannot be fully covered, however I find that this review would be of potential interest. The authors have done a very good job, especially with regard to the recent ML methodology employed in molecular docking and this should be emphasized if not being part of the title. I suggest publication of this review after some minor corrections and clarifications given below:

a)    If the first author is still affiliated to the University of Cambridge, his academic e-mail address should be used instead of a Gmail account.
b)    “exploit” is frequently used with a negative meaning – used “employ” instead
c)    You could add an equation in the second paragraph of p. 3 to describe the general form of the energy terms used, including the desolvation and internal ligand energy.
d)    The part of the sentence “via a molecular mechanics force field paradigm” in par. 3 of p. 3 does not make sense, could be erased.
e)    The statement “AutoDock Vina does not use atomic charges at all” end of p. 3 to top of p. 4 is misleading rather than informative of how Vina uses a different treatment for the electrostatics.
f)    Par. 2 in p. 4 is a good place to add a comment with refs. about the effect of the number of ligand degrees of freedom in the predictive strength of molecular docking.
g)    The rest of the manuscript is well-structured and concise, however, I suggest that proof reading from a native English speaker would have a great impact to the final form of the review.

Author Response

a) If the first author is still affiliated to the University of Cambridge, his academic e-mail address should be used instead of a Gmail account.

Changed as requested.

b) “exploit” is frequently used with a negative meaning – used “employ” instead

Changed as requested.

c) You could add an equation in the second paragraph of p. 3 to describe the general form of the energy terms used, including the desolvation and internal ligand energy.

We have included a general equation derived from the MMFF94 force field as per reviewer’s request. Besides, ee decided to change the classification of AutoDock score function to empirical. Even it has terms originated from force field, it calculates coefficients empirically through linear regression analysis.

d)The part of the sentence “via a molecular mechanics force field paradigm” in par. 3 of p. 3 does not make sense, could be erased.

The sentence was changed to Charges are usually assigned through algorithms that distribute the net charge of a molecule among its constituent atoms as partial atom-centered charges.”.

e)    The statement “AutoDock Vina does not use atomic charges at all” end of p. 3 to top of p. 4 is misleading rather than informative of how Vina uses a different treatment for the electrostatics.

We changed the sentence to: “AutoDock uses Gasteiger-Marsili atomic charges whereas the closely-related AutoDock Vina does not require the assignment of atomic charges, since the terms that compose its scoring function are charge-independent [3,24]⁠.

f)Par. 2 in p. 4 is a good place to add a comment with refs. about the effect of the number of ligand degrees of freedom in the predictive strength of molecular docking.

We have added the following paragraph in the section pointed out by the referee: "The overall size of the ligand, especially if it contains a large number of rotatable bonds impacts most docking algorithms in a negative way, both in terms of computational cost of each individual docking run and in terms of docking accuracy [49]. That is the case because each new rotatable bond inherently increases the ligand's degrees of freedom, thus increasing the number of possible conformations. The enhanced conformational space is therefore much more complex to explore, rendering less accurate results, usually even with increased sampling steps. The magnitude of this effect is distinct for different algorithms [3,50] and fragment-based ones seem to exhibit superior performance in such cases [49]."

g)The rest of the manuscript is well-structured and concise, however, I suggest that proofreading from a native English speaker would have a great impact to the final form of the review.

We have thoroughly reviewed the manuscript and we believe it is now much improved in this aspect.

Reviewer 3 Report

The review titled "Key topics in molecular docking for drug design" is well presented. The visual representation of data is lacking. Overall, this review could be suitable for the journal

Author Response

The review titled "Key topics in molecular docking for drug design" is well presented. The visual representation of data is lacking. Overall, this review could be suitable for the journal.

We added an additional table containing the mostly used docking software to facilitate the overall accessibility of some of the elements in the text.

Round 2

Reviewer 1 Report

The authors have performed satisfactory all the suggestions I made during my first revision round.

I have only a minor suggestion for them. Unfortunately, eHiTS is not currently available since Simbiosys was bought in 2014 by John Wiley & Sons. Then, I suggest to remove it from Table 1.

Congratulations to the authors for their excellent work

Author Response

We thank once more the valuable reviewer suggestions. We have accepted the final suggestion and modified table 1 accordingly. Besides, we took the chance to revise the manuscript again and added a more detailed description of equation 1. All changes are marked in red.